# Effects of Short Practice of Climbing on Barriers Self-Efficacy within a Physical Education and Sport Intervention in Germany

**DOI:** 10.3390/sports7040081

**Published:** 2019-04-04

**Authors:** Mirko Krüger, Christiane Seng

**Affiliations:** 1Division of Social and Behavioral Sport Sciences, University of Duisburg-Essen, 45141 Essen, Germany; 2Division of Sport and Education, University of Osnabrück, 49080 Osnabrück, Germany; christiane.seng@uni-osnabrueck.de

**Keywords:** barriers self-efficacy, indoor wall climbing, belaying, physical education and sport, field experiment, physical activity, Germany

## Abstract

The study examined the effects of an indoor wall climbing intervention within the context of a regular Physical Education and Sport (PES) program on barriers self-efficacy (SE) of adolescents in Germany. The study used a field experiment with a wait-list control group. Seventy-eight 8th-graders were included (age: 14.41 ± 0.71 years), with 37 randomly assigned for the intervention group and 41 for the control group. The intervention group participated in two half-day indoor wall climbing excursions (duration: 180 min each) based on SE building strategies. Both groups were pre-and post-tested in SE of indoor wall climbing and belaying. The control group did not receive any treatment before post-test. After the intervention, significant improvements were found in the experimental group on SE of belaying (*F*(1,76) = 23.45, *p* = 0.000, *η*2*p* = 0.24) using repeated-measures ANOVA. This study provides the first evidence from a German PES field experiment on increasing an important SE facet related to indoor wall climbing among 8th-graders. The program may be improved and further analyzed to install a short-term method to achieve one important educational goal within ordinary PES programs in Germany and to contribute to the personal development of the students.

## 1. Introduction

Supporting a healthy lifestyle and involvement in sport throughout life and promoting students’ psychological, social, and personal development are two major objectives of Physical Education and Sport (PES) programs worldwide [1,2]. Still, the foci may differ between countries due to different sport–pedagogically formed conceptualizations of both objectives. Against this background, PES programs have been found to have physical, mental, and psychosocial benefits [3,4,5,6,7,8]. At the same time, the current evidence is inconsistent, and we do not fully understand the underlying mechanisms of these effects within specific and sport culturally diverse PES settings [3,9,10]. This knowledge gap challenges the development of effective interventions within the context of regular PES programs, also in Germany.

According to Garn [11], one important psychological outcome of quality PES programs is self-efficacy (SE). It is defined as “people’s judgement of their capabilities to organize and execute courses of actions required to attain designated types of performances” ([12], pp. 391) and was found to mediate the relationship between PES and the students’ physical activity (PA) [10]. A few studies indicate that PES influences global or PA-related SE [10,13]. A systematic review perceives SE as an important psychological construct to focus on in future PA interventions [14].

SE is one of the basic concepts of determinants and considered as the primary agent in Banduras’ [15] social cognitive theory (SCT). Related to PA or exercise, SE is often interpreted as barriers self-efficacy (the perceived ability to perform a behavior in the face of barriers, e.g., “I am able to exercise outside when the weather is bad”), task-specific self-efficacy (the perceived ability to perform a particular behavior, e.g., “I am able to exercise twice a week”), or as adherence self-efficacy (the perceived ability to adhere to a specific regimen of behavior, e.g., “I am able to continue exercising twice a week for 30 min at moderate intensity without quitting all next year”) [16]. Within the SCT framework, (exercise) behavior is seen as an outcome of a dynamic and reciprocal interaction between influences by person, behavior, and environment [12]. Through purposeful and goal-directed actions, individuals believe that they can influence their own behavior and the surrounding environment [17]. Ultimately, this influences their future behavior. Bandura [18] identifies four information sources to affect SE: Mastery experience as the most influential source (i.e., successful performance of the target behavior), vicarious experience (i.e., seeing a ‘role model’ successfully perform the behavior and evaluating one’s own performance against the performance of that social model), verbal persuasion (i.e., others express faith in the individual’s capability to master specific behaviors), and physiological or emotional states (i.e., regulating emotional states and correcting misinterpretations of physiological states).

The hierarchical and multifaceted exercise and self-esteem (EXSE) model proposed by Sonstroem, Harlow, and Josephs [19] suggests that specific behavioral self-efficacies influence general behavioral competencies [19]. Therefore, SE can be regarded as a multilevel and multifaceted set of beliefs, each differing in strength, and generativity [20]. In line with that, general SE measures have not been as useful in predicting people’s behavior under more specific circumstances as more specific SE measures [21]. This points to the fact that an individual could have a high level of, e.g., PA-related general SE but maybe low SE in concrete performance situations (e.g., climbing, running, playing basketball), because each type of PA consists of a specific set of behaviors, performed in a specific context. Consequently, to detect the effects of specific PES programs, it is necessary to use satisfactorily specific SE instruments [16].

Nevertheless, most previous PES studies regarding SE use general SE measures or, in a more global sense, PA-related SE scales that were not explicitly focusing on specific behavioral SE, e.g., [22,23,24,25]. Therefore, we cannot exactly estimate the intervention benefits regarding specific SE change related to the exercise tasks in the studies. By contrast, the study from Lubans et al. [26] analyzed the psychological effects of a free weights and elastic tubing resistance training in adolescents with the help of a task-specific resistance training SE scale.

Against this background, climbing is discussed and proven as one important PES topic with benefits for students’ health-related fitness (e.g., muscle-strength and endurance) and an influence on psychosocial correlates.

Firstly, (indoor wall) climbing is extremely popular [27,28]. Secondly, it is a fundamental form of exercise from early childhood on and challenges children and adolescents at all levels of expertise, functionally and cognitively [29,30]. Further, it requires a multifaceted repertoire of movements, demands strength, coordination, cardiorespiratory fitness, and mental health [31,32,33,34,35]. Evidence reviewed by Siegel and Fryer [36] suggests that climbing in youth may also increase psychological achievements (e.g., self-efficacy). Additionally, indoor wall climbing has the benefit of removing several outdoor climbing risk factors (e.g., falling rocks). It was found to be the safest option for practicing rock-climbing [37].

However, research on the benefits of wall climbing for children within regular PES programs is scarce and frequently based on anecdotal evidence [36]. In particular, there is a lack of studies focusing on psychological benefits (e.g., barriers SE of climbing). Still, Mazzoni et al. [38] did find improvements of participants’ SE regarding climbing and belaying in a six-week indoor wall climbing intervention for children with special needs aged 6–12 years. Stoll et al. [39] point out the potential of using climbing in PES contexts as opposed to sports activities without social support. Their study results indicate higher general SE values for people who participated in a three-month climbing course compared to people participating in an aerobic fitness course. Thus, the available evidence suggests positive outcomes, which is in line with research on the impact of climbing and rope courses on SE among various samples (e.g., depressive participants) outside school and regular PES programs and using different SE building strategies [35,40,41,42,43,44,45,46,47].

No studies have yet attempted to measure the effects of indoor wall climbing on barriers SE within a PES intervention in Germany, although this is aligned with school-based educational goals regarding personal development inside regular PES programs [48,49,50]. Therefore, the study presented here aims to close this research gap and examine this specific issue. Specifically, we were interested in the impact of the short practice of indoor wall climbing on students’ barriers SE of climbing and belaying as basic tasks. The assessment of SE judgments followed the EXSE model proposed by Sonstroem et al. [19]. We hypothesized that successful climbing experiences during the short-term intervention would affect SE on climbing and belaying especially and contribute to the development of the students’ selves, as supposed by the findings of several brief SCT-based interventions, e.g., [51]. This is one important education goal of regular PES programs in Germany [48] and may influence future exercise behavior and its benefits [52,53].

## 2. Materials and Methods

### 2.1. Participants

The participants (8th-graders) were chosen from one secondary school (ISCED 3A level) in the city of Essen (North Rhine-Westphalia, Germany). They were not aware of the specific purpose of this study. In total, 78 young adolescents were randomly assigned to either an intervention group (IG; *n* = 37; 18 males, 19 females; mean age 14.36 ± 0.59 years) or a control group (CG; *n* = 41; 16 males, 25 females; mean age 14.45 ± 0.81 years), with similar distributions between both groups regarding gender (*χ2*(1) = 0.02, *p* = 0.964) and migration background (*χ2*(1) = 0.73, *p* = 0.391). The multivariate analysis of variance (MANOVA) results reflected no statistically significant differences at multivariate level (*F*(8,67) = 0.90, *p* = 0.578) for any of the investigated independent variables (see Table 1).

The study was conducted in accordance with the declaration of Helsinki [54] and approved by the ethics committee of the Medical Faculty of the University of Duisburg-Essen (No. 17-7738-BO). All parents and students provided their written informed consent before taking part in the study.

### 2.2. Procedure

The intervention took place in August 2017, at the beginning of the school year, and was announced as a regular PES program for the adolescents. It consisted of two half-day excursions in the morning to a climbing gym (duration: 180 min each). The intervention group climbed in August. The wait-list control group did not receive any treatment before the post-test and climbed in September after the pre- and post-tests. Based on the main sources of influence on SE [18], the intervention goal was to create an environment in which the adolescents could (a) have lots of successful climbing activities, (b) see peers perform successfully, (c) experience verbal persuasion by peers and/or adults (staff and/or teachers), and (d) regulate emotional states to strengthen their barriers SE of climbing and belaying. Because of the explorative study character and in order to maintain a realistic climbing environment in the climbing gym, we consciously opted against regulating the specific impact of each of the SE sources within the intervention. However, we assumed mastery experiences as the most influential source as it is suggested to be particularly influential within climbing contexts [38,55]. All participants experienced the same amount of time climbing and belaying during the two half-day interventions.

A trainee teacher with a bachelor’s degree in PE supervised the intervention. Each half-day excursion started with a short briefing on the general organization of the day (the time frame, working in pairs, the counterparts, safety rules before, during, and after climbing) and relevant climbing techniques (nodes, belaying). In that, we followed the recommendations for indoor wall climbing with school classes [29,56]. During the entire intervention period, professional staff of the climbing gym monitored the adolescents in the background. When necessary, they helped physically to facilitate students’ mastery experiences [18], e.g., by encouraging them to be courageous when choosing an appropriately demanding level. Staff were not aware of the specific purpose of this study. Following an inductive and student-centered learning approach to promote SE [20], the adolescents could freely choose their belaying partner and their individual difficulty level for the climbing routes. According to recent study results, “young children are relatively sensitive enough to their action boundaries for climbing and, therefore, may be able to make informed decisions themselves about whether a surface is climbable” ([57], p. 134).

To ensure the accuracy and precision of the intervention, the two class teachers were also present. The trainee teacher took notes immediately after each half-day excursion to document positive or negative saliencies during the intervention period regarding students’ and staff behavior and social interaction. The intervention group evaluated both half-day excursions during the post-test on a 7-point semantical differential (Question: “How did you experience the climbing excursions?”) as interesting (1.70 ± 0.93; unit of measurement: 1 = interesting to 7 = not interesting), successful (2.14 ± 1.11; unit of measurement: 1 = successful to 7 = not successful), and demanding (2.35 ± 1.13; unit of measurement: 1 = demanding to 7 = not demanding), indicating that the intervention was realized as intended in order to promote positive change in barriers SE of climbing and belaying.

The data were recorded using a standardized questionnaire filled in by the adolescents with the help of the trainee teacher and the two class teachers. Filling in this questionnaire took approximately 15 min (pre-test) and 10 min (post-test).

### 2.3. Measures

Students’ barriers SE of climbing and belaying was measured by two newly developed SE scales, as such measures are missing for specific PES contexts [58]. Scales from other studies did not correspond to our research interest, e.g., [38]. The item development process involved two writers: One of the writers is an established faculty member in teacher education, specialized in PES. The second writer is a Master’s student with the same expertise. The original item pool that could be integrated into Banduras’ [20] framework was constructed from a comprehensive literature review on SE and climbing. For this, we retrieved papers, books, and other relevant literature (e.g., recommendations by school administrations), and explanations for relevant aspects of climbing and promoting barriers SE in PES programs [29,56,59,60]. In several sessions, we drew various task-related, social, cognitive, physiological, and emotional state key barriers from the literature and conducted several rounds of internal in-depth deliberations on the items.

Based on the conceptualization of Bandura [20] and the key barriers mentioned, two other experts, teaching and researching in the areas of PES and indoor wall climbing, discussed, edited, and revised the initial pool of items from the item development for content and face validity as well as for clarity and conciseness in several meetings with both item developers. They were provided with the purpose of the measure, definitions of the key barriers, and the underlying framework. Moreover, they were asked to provide any additional comments on how to refine the items. In the meetings, we reached consensus on reducing the initial pool of items. During the process, 3 items were removed due to weak relevance. The content validation phase resulted in a 20-item instrument. After that, two adolescents were asked to complete this version of the questionnaire. They responded to the items with a common stem for both scales “Please indicate how sure you are to perform the described situation”, indicating their level of disagreement or agreement on a scale from 0 to 100 in 10-point increments (ranging from 0% = surely not to 100% = most sure), while evaluating the clarity of the items and providing feedback on the response scales at the same time. The feedback suggested slight changes to 3 items. A full list of the items translated into English for better readability and the original questionnaire in German are attached (see Appendix A).

According to Loewenthal [61], reliability tests showed good internal consistency for both scales, SE of climbing (Cronbach’s *α* [t1] = 0.92; Cronbach’s *α* [t2] = 0.95) and SE of belaying (Cronbach’s *α* [t1] = 0.92; Cronbach’s *α* [t2] = 0.95). Test–retest reliability was *r* = 0.82 (*p* < 0.00) for SE of climbing and *r* = 0.70 (*p* < 0.00) for SE of belaying. Concurrent validity was analyzed for both scales using a validated 3-item scale measuring global SE (ASKU) as an external criterion [62]. We expected moderate significant correlations by comparing ASKU with both barriers SE scales, which were confirmed with *r* = 0.54 (*p* < 0.00) for SE of climbing and *r* = 0.44 (*p* < 0.00) for SE of belaying. The means of both scales in the pre- and post-tests were used to assess the intervention effect.

Further data (age, gender, migration background, self-reported sportiness, relevance of sport) were integrated into the questionnaire using items from the most noted PES study in Germany [63]. Frequency and quality of prior climbing experiences was assessed by two items.

### 2.4. Statistical Analysis

All data were screened for missing data and outliers: Across both sub-samples (intervention and control group), none of the variables had missing values.

A baseline check (MANOVA) was conducted to analyze the baseline equivalence between both conditions regarding SE of climbing and SE of belaying, the two dependent variables. The main analysis testing the intervention effect was performed using a series of 2 (Group: IG vs. CG) × 2 (Time: Pre-test vs. post-test) repeated-measures analysis of covariance (ANCOVA) with gender as covariate. We specifically looked at the two-way interaction of Group × Time and the three-way interaction of Group × Time × Gender. Significant interactions would indicate different changes regarding self-efficacy on indoor wall climbing and belaying between IG and CG.

Additionally, partial eta square (*η*2*p*) was calculated to determine whether a statistical difference was practically meaningful. A value of 0.01 ≤ *η*2*p* < 0.059 represents a small effect, 0.059 ≤ *η*2*p* ≤ 0.138 represents a medium effect, and *η*2*p* > 0.138 represents a large effect [64]. All analyses were performed using the Statistical Package for Social Sciences (SPSS) Version 23.0 (IBM, Chicago, IL, USA), and significance level was set at *p* ≤ 0.05.

## 3. Results

The MANOVA results reflected no statistically significant differences at multivariate level (*F*(2,75) = 0.01, *p* = 0.558, *η*2*p* = 0.01) for any of the investigated dependent variables. Thus, both groups did not differ in SE of climbing (IG: 57.82 ± 20.97; CG: 63.14 ± 21.92) and SE of belaying (IG: 58.75 ± 22.24; CG: 62.40 ± 25.17) at pre-test. Their initial SE on both tasks, climbing and belaying, was comparable.

There was no significant group by time effect (*F*(1,76) = 1.84; ns) and no group by time by gender effect (*F*(1,76) = 0.00; ns) for SE of climbing (see Table 2). However, it is worth noting that, while SE of climbing remained rather the same among CG adolescents (pre-test: 63.82 ± 22.07 and post-test: 64.61 ± 23.76), IG students expressed a slightly positive change in SE of climbing (pre-test: 57.11 ± 21.12 and post-test: 62.88 ± 25.98) over time. The ANCOVA yielded a medium main effect for time (*F*(1,76) = 5.58, *p* = 0.021, *η*2*p* = 0.07) and a small main effect for gender (*F*(1,76) = 4.52, *p* = 0.037, *η*2*p* = 0.05), but not for group (*F*(1,76) = 1.77; ns).

The repeated measures ANCOVA for SE of belaying revealed a significant group by time effect (*F*(1,76) = 23.45, *p* = 0.000, *η*2*p* = 0.24). The effect is large (see Table 2). The IG adolescents showed an increase in SE of belaying over time (pre-test: 57.87 ± 22.57 and post-test: 74.40 ± 22.08). On the other hand, CG adolescents seemed to show similar patterns, as their SE of belaying slightly decreased over time (pre-test: 62.53 ± 24.86 and post-test: 60.70 ± 27.70). The ANCOVA produced a large main effect for time (*F*(1,76) = 15.42, *p* = 0.000, *η*2*p* = 0.17), but not for group (*F*(1,76) = 0.75; ns) and gender (*F*(1,76) = 2.11; ns). No significant group by time by gender effect was found (*F*(1,76) = 0.00; ns).

## 4. Discussion

The aim of the present study was to examine the effects of indoor wall climbing on barriers SE of climbing and belaying within a PES intervention for 8th-graders using a field experiment that referred to an established theoretical framework and included a control group. To the best of our knowledge, no other study has yet focused on this particular issue within regular German PES programs. Consistent with the EXSE model [19] and SCT [12], our hypothesis postulated that the PES intervention would lead to higher improvements regarding barriers SE of climbing and belaying within the context of two half-day excursions in a climbing gym. The hypothesis was partly confirmed in accuracy measures of SE of belaying where the intervention group showed higher values than the control group. No significant effect was found for SE of climbing. This suggests that the intervention improved SE for the target group. These findings are in line with earlier findings on climbing and rope courses within and outside of PES programs obtained from a sample of students with special needs [38], psychotherapy patients [40], young adults taking part in a recreational therapy [43], childhood cancer survivors [46], and people with depression [35].

The missing effect on barriers SE of climbing may be due to the short duration of the intervention. One hint for this assumption is the fact that SE of climbing remained rather the same among CG adolescents, while IG students expressed a slightly positive change in SE of climbing over time, which could have become significant by adding more climbing time for mastery experiences. With reference to Kruger and Dunning [65], another reason for the missing effect on barriers SE of climbing could lie in a cognitive bias of the adolescents leading to more unrealistic (i.e., higher) SE values at pre-test regarding their own climbing skills and a more realistic SE value at post-test after experiencing their actual skills (‘Dunning-Kruger effect’). This could narrow the pre–post-test values gap and maybe limit the intervention effect for barriers SE of climbing.

The intervention proved to be sensitive on barriers SE of belaying as a probably less familiar task compared to the climbing task. Literature shows that learning unknown tasks bears more potential for learning progression and positive SE changes [66]. Feedback from the supervisor regarding the novelty of the task and the behavior of the adolescents during the intervention period supports this reflection. However, we cannot clearly confirm the assumption, as we only collected data for prior climbing experiences without differentiating between climbing and belaying tasks. Therefore, future PES studies should design climbing interventions to test the supposed time and novelty effects in more depth. Previous findings within the context of climbing interventions inside and outside PES programs suggest positive effects using longer intervention periods, e.g., [35,38,39,40,67]. Such intervention formats could not be realized in regular PES programs in Germany but maybe in additional ones.

### 4.1. Study Limitation

Our findings suggest that for this group of adolescents (8th-graders), the PES climbing intervention has short-term effects on barriers SE of belaying, indicating that it can contribute to the development of the students’ selves. This is one important education goal of regular PES programs in Germany [48,49,50]. Nevertheless, the knowledge base for that specific topic remains sparse, as the current study is the first to analyze such effects within the context of regular German PES programs. We do not know about the stability, generalization, and practical significance of the effect, or which implications this may have for the adolescents regarding their mental and physical health and future exercise behavior [52,53]. Prospectively, the additional use of qualitative interviews could help to shed more light on the issue. Against that background, future studies should integrate the analysis of long-term effects using follow-up measures. This may help to determine if such effects can contribute to increased physical activity and improved global or more general PA-related self-efficacy and self-esteem, as proposed by the EXSE model [19].

Furthermore, it should be tested to what extent different PES intervention designs and techniques (e.g., varying in sources used to change SE or age) lead to different intervention effects. This can be expected from literature on the determinants of change in PA in children and adolescents [14], and findings on the specific impact of SE building strategies within climbing interventions outside PES contexts regarding mastery experience [38,43], vicarious experiences [41,42], verbal persuasion [43,44,45], and physiological or emotional states [47]. Would, e.g., more encouragement and social support from peers and/or staff to try and accomplish more difficult routes lead to more successful exercise behavior and significant positive SE changes regarding climbing and belaying? Similarly, physiological outcomes (e.g., PA, muscle strength, and endurance) could be included in time-extended interventions (e.g., two sessions per week for one month) using objective measures, as proposed by Siegel and Fryer [36]. That would strengthen the evidence of benefits of climbing within PES programs as realized in several studies outside PES [68]. In that, the pubertal stage of participants should be integrated that we did not assess in our study.

Barriers SE of climbing and belaying were developed for the current study. Both scales demonstrated good internal consistency, test–retest reliability, content, and concurrent validity. They may be used in future studies designed to explore adolescents’ barriers SE related to climbing and belaying tasks. In addition, future studies should test their factorial validity and extend the developed measures for task-specific and adherence SE facets within climbing contexts.

Albeit preliminary, the results of the present study suggest specific effects of the two half-day regular PES interventions at secondary school level on barriers SE of belaying. To our knowledge, this is the first PES study to examine specific SE in climbing situations in adolescents in Germany. Still, the findings should be treated with care as they are based on a relatively small sample from one school. The study may, in fact, have been underpowered, especially to detect small between-group differences regarding SE of climbing. Future studies should include a sample size estimation and power analysis prior to the intervention implementation [69,70].

### 4.2. Study Strong Points

The results encourage further studies with larger randomized and controlled trials to test the replicability of effects as well as to extend the current study with further dependent variables measuring important physical, mental, and psychosocial outcomes within regular PES programs [3,71].

## Figures and Tables

**Table 1 sports-07-00081-t001:** Pre-test characteristics of the participants by condition.

Characteristic	IG (*n* = 37)	CG (*n* = 41)
Gender = female (%)	52.8	62.5
Migration background = yes (%)	29.7	29.3
Age, years (mean ± SD)	14.36 ± 0.59	14.45 ± 0.81
Self-reported sportiness ^a^ (mean ± SD)	3.46 ± 1.23	3.51 ± 1.16
Relevance of sport: school-related ^b^ (mean ± SD)	3.92 ± 0.92	4.07 ± 0.95
Relevance of sport: club-related ^b^ (mean ± SD)	3.11 ± 1.46	3.08 ± 1.49
Relevance of sport: non-formal ^b^ (mean ± SD)	3.41 ± 1.23	3.58 ± 1.48
Frequency of climbing experiences ^c^ (mean ± SD)	2.95 ± 0.81	3.27 ± 1.00
Quality of climbing experiences ^c^ (mean ± SD)	3.92 ± 1.29	3.98 ± 1.23
Global self-efficacy ^d^ (mean ± SD)	3.52 ± 0.87	3.77 ± 0.80

Notes: ^a^: Score range (SR) from 1 = very unathletic to 5 = very athletic; ^b^: SR from 1 = not important to 5 = very important; ^c^: SR from 1 = never to 5 = often; ^d^: SR from 1 = not at all to 5 = totally agree.

**Table 2 sports-07-00081-t002:** Analysis of covariance results for barriers self-efficacy (SE) of climbing and belaying.

Effects			*F*	*df*	*p*	*η*2*p*
Barriers SE of Climbing	Main	Time	5.58	1	0.021	0.07
		Group	1.77	1	0.187	0.02
		Gender	4.52	1	0.037	0.05
	Interaction	Time × Group	1.84	1	0.179	0.02
		Time × Gender	3.42	1	0.068	0.04
		Gender × Group	2.63	1	0.109	0.03
		Time × Group × Gender	0.00	1	0.945	0.00
Barriers SE of Belaying	Main	Time	15.42	1	0.000	0.17
		Group	0.25	1	0.615	0.00
		Gender	2.11	1	0.150	0.02
	Interaction	Time × Group	23.45	1	0.000	0.24
		Time × Gender	1.94	1	0.168	0.02
		Gender × Group	1.54	1	0.218	0.02
		Time × Group × Gender	1.76	1	0.188	0.02

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
