# Peer review of "Effects of Short Practice of Climbing on Barriers Self-Efficacy within a Physical Education and Sport Intervention in Germany"

_sports, 2019, doi:10.3390/sports7040081_

Round 1
Reviewer 1 Report
General comments
This study investigated the possible effect of an indoor wall climbing intervention on barriers self-efficacy of adolescents. To do this the Authors created a questionnaire to assess pre- and post-intervention self-efficacy of indoor wall climbing and belaying. The study seems rather interesting; however, as clarified by the specific comments below some points should be addressed more extensively.
Specific comments
Language:Improve the scientific English for clarity.
Introduction
Page 1. Lines 38-38. The references Garn [12] and self-efficacy (SE) [11] do not correspond to the final list of references.
Page 1 line 40. “….to attain designated types of performances” (p. 391). Please, add the reference.
Page 2 line 42. “A recent systematic review perceives SE…”. Please, delete the term recent.
Participants
Table 1 is not clear. Please add the unit of measure of the measured parameters and specify what parameters are expressed as mean ± SD or as %.
Procedure
No reference is made on the control group. I think that the authors need to describe what the control group students did during the intervention. Moreover, if the control group did not climb, why Authors tested the pre- and post-intervention self-efficacy of indoor wall climbing and belaying of the control group? It is not clear. Please, clarify.
Page 3 lines 128-129. Please delete the sentence “Adolescents were randomly assigned”.
Page 4 lines 153-156. What do the numbers 1.70 ± 0.93; 2.14 ± 1.11; 2.35 ± 1.13 refer to? If they are results of the questionnaire, these data should be reported in the results section. Moreover, add the unit of measure.
Results
Please, add the unit of measure through the results section.
Please, delete the Figure 1(a) since the results were not statistically significant. In the Figure 2(b) add the standard deviation and the significance (* in figure and p = …. In legend). Page 6 line 232 and line 234. Please delete the data referring to the SE of belaying over time of IG and CG. These data are already showed in figure 1(b).
References
Please, translate the references in German and French into English language (e.g. Flecken, G.; Heise-Flecken, D. Klettern in der Halle. [Title translated]. Meyer & Meyer: Aachen: Germany, 2015.
Author Response
I want to thank you for your constructive and helpful feedback towards the above-mentioned manuscript. The uploaded file contains all of your comments and how I did handle them during the revision process of the manuscript. Formal comments (e.g., rearranging not corresponding references, deletion/insertion of words, adding of references and units of measurements, translation of literature) are not mentioned. I did include them all in the revised version.

Reviewer 2 Report
The purpose of this study was to assess the effects of indoor wall climbing on barriers Self Efficacy (SE) in German adolescents. The study was conducted during a Physical Education and Sport (PES) program. Participants were divided into an intervention group (two half day indoor wall climbing) and a wait list control group (subjects did not receive any treatment before post-test). The results evidenced that the experimental group showed significant improvement only on SE of belaying. Despite some positive aspects of the study, there are some concerns regarding weakness of detail in the methods paragraph, the interpretation and analysis of data and, last the conclusions drawn. Indeed, study’s procedures and methodologies are hard to follow and understand. There are a number of concerns that need addressing in order to achieve sufficient quality for publication
General Comments:
The author explained that the intervention was announced as a regular PES program, however it consisted in only two half day excursions. Therefore, I think that it is hard to consider it as a PES intervention. For this reason the research design does not seems appropriate according to the title. In addition, as suggested by the author, the short duration of the intervention could be a relevant limit of the present study which could have caused the missing effects on barriers SE of climbing. I suggest to change the title according to the procedure (e.g. effects of short practice of climbing on … or acute effects of…).
- In the introduction paragraph, it is not clear why the authors supposed that two practice of climbing might improve subjects’ SE. Please add the rationale of the study in the introduction paragraph, also adding a scientific English literature.
- Methods are not adequately described and therefore the research is difficult to replay, a deeply revise is needed:
· More details are required on how the intervention was conducted, in particular it is not clear if all participants experienced the same amount of time climbing and belaying during the two half day interventions.
· It is not clear why the intervention group evaluated both half-day excursions on a 7-point semantical differential question (“How did you experience the climbing excursions?”) during the pre-test. Please explain that.
· The procedure used to develop the two SE scales are confused and hard to understand.
· Table 1 could have been a useful method to have a general overview of participant’s characteristics in the pre-test. However, without explain in the methods paragraph the score range, it is impossible to understand the different student’s conditions. In addition, considering that the study measured barriers SE belaying, it would have been useful to collect students’ belaying experience before the intervention. Moreover, unit of measurements are missing (e.g score).
· Statistical analysis is not appropriate. Did you assessed the effects of gender on SE?
· Did you assessed the pubertal state of the subjects? This might strongly influence the results. Indeed, gender might influence the SE results and this should be taken into consideration. Please add these information.
· Post-hoc differences between groups for each time point should only be conducted and the means reported individually, if there was a significant interaction (time x gender x group). On the other hand, a reporting of all statistical main and interaction effects (with at least the reached p-values) for all ANOVA analyses would strengthen the credibility of the manuscript.
- Results: The results should be rewritten according to the new statistical approach.
- Discussion: As results, discussion should be rewritten according to the results also adding scientific English literature.
Please add two paragraphs entitled “study limitation” (eg. no pubertal state was assessed) and “study strong points” in the end of the discussion paragraph
Author Response

(The authors gave the same response as above.)

Round 2
Reviewer 2 Report
The authors have very seriously taken into account reviewers' remarks and improved their paper accordingly. Some parts have been thoroughly rewritten. I have no other concerns to address.